# From the Friend to the Foe—*Enterococcus faecalis* Diverse Impact on the Human Immune System

**DOI:** 10.3390/ijms25042422

**Published:** 2024-02-19

**Authors:** Agnieszka Daca, Tomasz Jarzembowski

**Affiliations:** 1Department of Physiopathology, Medical University of Gdańsk, 80-210 Gdańsk, Poland; 2Department of Microbiology, Medical University of Gdańsk, 80-210 Gdańsk, Poland; tjarzembowski@gumed.edu.pl

**Keywords:** *Enterococcus faecalis*, commensal, immune system, inflammation, infection

## Abstract

*Enterococcus faecalis* is a bacterium which accompanies us from the first days of our life. As a commensal it produces vitamins, metabolizes nutrients, and maintains intestinal pH. All of that happens in exchange for a niche to inhabit. It is not surprising then, that the bacterium was and is used as an element of many probiotics and its positive impact on the human immune system and the body in general is hard to ignore. This bacterium has also a dark side though. The plasticity and relative ease with which one acquires virulence traits, and the ability to hide from or even deceive and use the immune system to spread throughout the body make *E. faecalis* a more and more dangerous opponent. The statistics clearly show its increasing role, especially in the case of nosocomial infections. Here we present the summarization of current knowledge about *E. faecalis*, especially in the context of its relations with the human immune system.

## 1. Introduction

*Enterococcus faecalis* is a gram-positive, facultatively anaerobic bacteria. They are considered quite resilient strains because e.g., they can grow in a wide range of temperatures and pH. The successful culture can be maintained in temperatures between 5 and 50 °C and pH as high as 9.6 [1]. The temperature resistance is linked to the high content of lipids and fatty acids in the membrane. It is especially important in low temperatures when the ratio of saturated and unsaturated fatty acids in the membrane is the most beneficial, but also to some degree, it maintains the resistance to higher temperatures [2]. The ability to survive in high pH and the presence of bile salts and enzymes allows them to colonize the small as well as the large intestine. It is believed that the tolerance of the wide variety of pH levels is linked with bacterial membrane structure, which is resistant to acids and alkalis, maybe due to the presence of H^+^-ATP pump activity [3]. The resistance to bile salts on the other hand comes from *E. faecalis*’s ability to produce bile salt hydrolases [4]. Another important ability of these strains allowing them to successfully survive in the intestine environment is their ability to form biofilm. The creation of the complex matrix, the barrier in a sense, in which they can function as a community adds to their resistance to the abovementioned factors [5].

## 2. *Enterococcus faecalis* as a Commensal

*E. faecalis* is considered a commensal in a human gastrointestinal (GI) tract, typically isolated from faeces in abundant amounts ranging from 10^5^ to 10^7^ bacteria per gram of stool [6,7]. It is also one of the first to colonize the human gut directly after birth. It takes 7–10 days to colonize the lower GI tract with *E. faecalis* post-partum [8].

Despite being one of the first to colonise the human GI tract, *Enterococcus faecalis* corresponds roughly to about 1% of the GI tract microbiome of an adult [9]. Its’ commensal nature is facilitated by high plasticity and the ability to adapt to an ever-changing environment. Even though they are not the most abundant bacteria of the adults’ intestinal microbiota, *Enterococcus faecalis* play an important role in maintaining the homeostasis of the GI tract by regulating the intestinal pH, production of vitamins and metabolization of nutrients such as carbohydrates, lipids, proteins and sugars [10,11]. They also take part in the elimination of pathogenic bacteria from the milieu of the intestines protecting the human body from the development of a wide variety of infections and inflammatory reactions [10]. Their tolerogenic nature permits tight cooperation with the immune system present in the GI tract.

It should be mentioned that even though the GI tract is the main site of *E. faecalis* habitation, it is considered a commensal also in other parts of the human body, such as the genitourinary tract (especially the vaginal tract), oral cavity and the skin [12].

### Immune System Cells Allow the Maintenance of the Commensal Nature of Enterococcus faecalis in the Gut

As mentioned above, *E. faecalis* plays an important role in maintaining the homeostasis in GI tract microbiome. It is partially possible thanks to the cooperation with the mucus (in this case gut) associated lymphoid tissue (GALT). GALT is considered the largest part of the human immune system because it covers the area of the whole GI tract—roughly about 260–300 m^2^ [13]. Even though it is a general belief that all mucus-associated lymphoid tissues (MALT) play a far more important role in human immunity than the so-called ‘classical’ immune system, the knowledge about their functioning is not fully elucidated. Partially it is due to their enormously diverse nature both in one person and in the whole human population. The enormity of the MALT itself is one element responsible for its heterogeneous nature, the other is the presence in many parts of the body with various properties, functions, daily exposure to outside danger and properties of invading pathogens. What is a common feature of MALT is that it needs contact with commensal bacteria to fully develop. In the case of GALT which development begins in the prenatal period, contact with bacteria starting to colonize the GI tract directly after birth is a crucial push to reach full maturity. The proof for that comes from germ-free animals in which mucus-associated lymphoid tissue never reaches full functional and structural capacity [14,15].

As the pre-sets, the properties of the GALT are tolerogenic towards bacteria residing in the intestines. However, it still needs to have a fine set of mechanisms able to react against e.g., pathogenic bacteria producing toxins or even commensal bacteria trying to cross the intestinal wall. One of the mechanisms impeding the invaders from spreading throughout the body system is commensal microorganisms themselves which apart from the basic competition for space and nutrients in the intestinal lumen can produce bacteriocins effectively eliminating the pathogens [16]. Additionally, the more robust the microbiome, the thicker and more impenetrable the mucus layer which protects the intestinal wall from the damage. What is more, the mucus layer protection is not only purely mechanical but also chemical, as it contains antibodies and antimicrobial molecules such as lactoferrin or lysozyme [17].

Another mechanism is purely immune-based. The GALT consists of inductive and effector sites. The first one is the site where the antigen is encountered, which activates the primary adaptive response. It consists of e.g., Peyer’s patches—the set of lymphocytes and antigen-presenting cells (APCs) enclosed in structured follicles and localized in or directly below the gut epithelium. The effector sites are the places where effector cells are sent. What is worth underlining, is that it means that the effector site does not have to be in the place of the immune response induction, such as the intestines. The cells activated at the inductive site can travel throughout the set of lymphatic and blood vessels towards any other accessible place, such as salivary glands, but the vagina or urinary tract as well [14]. That is another reason why the study of MALT is so difficult as mentioned above—its elements’ activation can affect not only the closest parts of the body but also the more distant ones.

The constant sampling of the microbial antigens from the lumen of the intestine by dendritic cells (DCs) and macrophages of inductive sites (e.g., aforementioned Peyer’s patches) allows the creation of ‘the library’ of the innocuous, generally ignored bacteria comprising physiological microflora. The sampling itself means the recognition of specific PRRs (pathogen-related receptors) by toll-like receptors (TLRs) of the immune cells, namely APCs. In the case of *Enterococcus faecalis*, the PRRs recognized typically by TLR2 are LTA (lipoteichoic acid) and/or lipoproteins. It activates anti-inflammatory, tolerogenic mechanisms, such as the production of anti-inflammatory cytokines, e.g., TGF-β (transforming growth factor β), and IL-10 (interleukin 10) and maintaining the tight junctions between the enterocytes [14,18,19]. The TLR9 on the other hand recognizes elements rich in CpG motifs. Even though CpG motifs are not abundant in *E. faecalis*, it is still believed that when bacteria manage to slip through the epithelial barrier, they can be recognized by TLR9 present on the basolateral surface of enterocytes and Paneth cells and by that any potential inflammatory reaction can be inhibited [14,20].

## 3. Probiotic Strains of *Enterococcus faecalis* and Their Impact on the Immune System

Due to its commensal nature, *E. faecalis* can be treated as a probiotic and in reality, there are food supplements rich in such strains with obvious immune-modulating properties. The first one manufactured has been used as long as for 70 years [21]. *E. faecalis* use as a probiotic is not as common as in the case of another enterococci—*Enterococcus faecium*, but still their presence on the supplements market has been established for many years now.

One of the most popular probiotic strains of *Enterococcus faecalis* is the DSM 16440 used in Symbioflor-1. The genomic analysis showed a lack of large parts of the enterococcal chromosome, mainly containing virulence-associated genes, including genes encoding cytolysin, enterococcal surface protein, and gelatinase. On the other hand, genes encoding aggregation substance, collagen adhesion protein, ensuring capsule formation and resistance to oxygen anions are still present, guaranteeing Symbioflor-1′s ability to colonize and survive in the intestinal milieu [22]. In the case of Symbioflor-1, a few conducted clinical trials assessed the impact of the bacteria on the development and management the allergy-related symptoms, mainly allergic rhinitis, chronic bronchitis and recurrent rhinosinusitis. The possible impact of bacteria on the protection against allergic reactions was proposed many years back in the so-called ‘hygiene theory’ by Strachan [23,24]. The exposure to bacteria switches the immune system response from the Th2 phenotype typical for allergic (and anti-parasitic) reaction into Th1 typical for an anti-bacterial and anti-viral response. It was hypothesized that the probiotic strain of *Enterococcus faecalis* would possibly play its role in the stimulation of the Th1 response and that would lead to the diminishment of the allergy severity of symptoms and their frequency. The effect of Symbioflor-1 is exerted mainly by stimulating the bacteria-host interaction in the GI tract. Therefore, the effect of DSM 16440 strain on the respiratory tract is achieved due to the stimulation of inductive sites in GALT and the propagation of the effect on other MALT elements—in their effector sites.

DSM 16440 strain, when supplemented for half a year by patients with chronic recurrent bronchitis lasting at least 2 years, caused the reduction of relapses by 43% during the supplementation, and by 68% during the 8 months after stopping the supplementation. Apart from that, in this double-blind, placebo-controlled study, the number of relapses and their severity were also visibly reduced when compared with the placebo group—the group of patients supplementing DSM 16440 did not need antibiotic therapy as frequently as placebo patients [25,26]. Apart from the switch towards Th1 response, the impact of Symbioflor-1 on the immune system is explained as the stimulation of IgA antibodies production. The data from animals showed that the amount of IgA antibodies in the saliva was significantly increased after the DSM 16440 strain supplementation. That in itself was caused by the stimulation of the plasma cells’ proliferation in the mesenteric lymph nodes [25,27]. IgAs are antibodies mainly existing in MALT, specifically in the mucus layer. Their ability to opsonize is much lower than classical IgGs. That in itself is another mechanism inhibiting unwanted acute inflammatory reactions in the GI tract and keeping its tolerogenic properties. The main function of IgAs is to agglutinate pathogens and to protect the gut wall from direct contact with bacteria. 

On the other hand, we have another *E. faecalis* strain—EF-2001, which probiotic properties are well-documented, but in contrast to *Symbioflor*-1, it is not used as a food supplement currently [28]. Its’ genomic structure shows a relatively low number of genetic factors, which may be linked with virulence potential, most of them related to capsule formation and adherence, where both those groups of genes are considered imperative in establishing probiotic host–bacteria interactions by inducing colonization and avoiding the removal from the host’s body [29]. The presence of antibiotic-resistance genes is linked with natural *Enterococcus* spp. antibiotic resistance to selected types of antibiotics, amongst them e.g., trimethoprim, and lincosamides. On the other hand, the EF-2001 strain contains relatively few genes involved in biofilm formation and genes encoding toxins ensuring its’ non-harmful properties [28]. 

Amongst the immunomodulatory properties studied the most for EF-2001 strain are those linked with the influence on the gut-brain axis and depressive disorders. The most established results come from olfactory bulbectomized mice (OBM) considered a valuable model for major depressive disorders [30]. The improper myelinati on in the prefrontal cortex (PFC) is found during the post-mortem assessment of depressed patients [31] and is therefore considered one of the processes leading to the development of depressive-like behaviour. Both brain-derived neurotrophic factor (BDNF) and leukaemia inhibitory factor (LIF) are involved in the process of myelination [32,33]. BDNF is decreased in the PFC of depressed patients as it was proved in post-mortem studies of depressed patients [34]. Takahashi and his team discovered that the treatment with EF-2001 strain enhances the CREB (cAMP response element-binding protein) expression and because BDNF is a target gene for CREB, therefore Akt/CREB/BDNF pathway induces the neuroplasticity and maturation of immature oligodendrocytes protecting them that way from abnormal differentiation observed in samples of depressed subjects [30]. The same team observed also that the EF-2001 strain is influencing the myelinization of the astrocytes. The increase in a LIF expression mediated by EF-2001 affected the levels of the myelin proteins such as MBP (myelin-based protein) and MAG (myelin-associated glycoprotein) and alleviated the depression symptoms by the enhancement of the NF-κB p65/LIF/STAT3 pathway [30]. The question remains though if the EF-2001 strain mode of action is in this case direct (through the vagus nerve) or indirect by inducing the specific changes in the properties of gut microbiota.

Other data suggest that the *E. faecalis* EF-2001 strain can also inhibit the lipids-mediated damage of the liver by the regulation of the AMPK (AMP-activated protein kinase) pathway [35]. It can also ameliorate the symptoms of atopic dermatitis by inhibiting the production of pro-inflammatory cytokines, e.g., TNF-α (tumour necrosis factor-alpha), and IFN-γ (interferon gamma) [36] and inhibit the development of acute gastritis by affecting the levels of pro-inflammatory response elements—by decreasing the inducible nitric oxide synthase (iNOS), cyclooxygenase-2 (COX-2), TNF-α, IL-1β and IL-6 [37]. But what should be underlined, the data are purely experimental and come from cell lines in vitro experimentation or from animal testing.

Other probiotic *E. faecalis* strains used in human supplementation are presented in Table 1.

The history of over 20 years of successful usage of *Enterococcus faecalis* strains as probiotics should not mask the dark side of the bacteria though. It is worth remembering that, the ease with which the bacteria can acquire various virulence factors and antibiotic resistance-related genes can pose serious problems, which need to be taken into consideration when planning its’ utilization as a probiotic. It is believed e.g., that plasmid-mediated genes in enterococci have had an impact on the development of Vancomycin-Resistant Enterococci (VRE), especially important in medical settings [39,52]. That is also partially the reason why the European Food Safety Authority (EFSA) did not put the *Enterococcus faecalis* on the list of Generally Recognised as Safe (GRAS). *Enterococcus* spp. as a whole genus is not included on the Qualified Presumption of Safety (QPS) list either [52,53]. 

## 4. *Enterococcus faecalis* and Dysbiosis

The balance in the host–microbiome interactions called eubiosis is very delicate. Any disturbances can lead to so-called dysbiosis development, which is a state of a given microbiome homeostasis damage. According to Petersen and Round, we can identify three types of dysbiosis. They can be caused by: the loss of beneficial microorganisms, the expansion of potentially harmful pathogens and the loss of microbial diversity [54]. The list of factors which can lead to dysbiosis development is quite impressive, among them the overall function of the digestive system (e.g., the amount and bactericidal properties of fluids produced by the liver and Paneth cells), the properties and the amount of IgAs produced by the plasma cells, diet (e.g., its content of carbohydrates and fibres), but also age, the level of physical activity and the presence of inflammatory reactions, even those not directly linked with the GI tract. But the most commonly mentioned factor facilitating dysbiosis development (all types of them) is antibiotics’ (over)consumption. That in itself is sufficient to disrupt the delicate balance in commensals inhabiting the human gut. It allows either the overgrowth with existing probiotic bacteria, which are resistant to currently used antibiotics and then due to their increased number they become pathogenic, or colonisation by pathogenic bacteria resistant to antibiotics [55,56].

As mentioned above *E. faecalis* is a very pliable bacterium, able to adapt to even fast changes in the local environment. Unfortunately, the feature which on one hand can grant the bacteria to survive the harsh realities of the ever-changing human GI tract can also allow for the easy transformation into pathogenic species. That is a reason why *E. faecalis* is sometimes called ‘pathobiont’—the bacteria with an overall positive influence on immunocompetent host systems functioning, but able to cause serious problems in those whose systems are somehow compromised and/or whose immune system cannot work properly [54]. 

In the case of commensal *E. faecalis*, according to Repoila et al., one of the elements controlling the bacteria in the gut is deoxycholate (DCA) [57]. DCA is a product of the de-conjugation of cholate (CA) facilitated by bacterial bile salt hydrolases, so its production is controlled in the colon by the eubiotic microflora itself [57,58]. As *Enterococcus faecalis* is sensitive to DCA, this substance can control the replicative and transcriptional activity of specific bacterial genes, including those implicated in the growth. That way the host’s body protects itself from bacterial overgrowth. Interestingly the same substance also stimulates genes participating in stress adaptation, such as genes ensuring the resistance to the bile salts (e.g., *gls24/B*), and antimicrobial peptides produced by either the other bacteria or the host (e.g., *gelE*, *sprE*), which ensures the survival of *Enterococcus faecalis* in the colonic lumen [57,59].

When the dysbiosis develops, the colonic step of CA de-conjugation does not happen. Instead, from CA and taurine taurocholate (TCA) is produced. In contrast to DCA, TCA stimulates mostly the genes involved in nucleotides (*ef0014* and *ef1719*) and amino acids (*ef0634/36* and *ef3106/10*) metabolism, promoting that way bacterial growth and leading to uncontrolled *Enterococcus faecalis* proliferation and overgrowth in the absence of inhibiting balance of DCA [57].

It is believed that a certain level of enterococci is required to attempt intestinal translocation so the first step for possible gut barrier crossing is the overgrowth of *E. faecalis* in the gut [60]. Not surprisingly, the most susceptible subjects are those immunocompromised. Successful maintaining of the gut barrier as protection from invasion by bacteria, as mentioned above, depends on the functional immune system and its cells. Among them are dendritic cells and macrophages which are continuously sampling the intestinal lumen for antigens. One of the phenotypical features of such macrophages is the expression of CX3CR1. The functional feature is their general anti-inflammatory activity, as they produce high quantities of e.g., IL-10, TREM-2 (triggering receptor expressed on myeloid cells 2), IL-23 and IL-1β. IL-23 and IL-1β in turn support the production of IL-22 by intraepithelial lymphoid cells (ILCs) [61,62]. Thanks to that the integrity of the gut wall can be maintained at the time when the healing of the damage is needed. They also contribute to the integrity of the gut wall through the supporting CD4+ T helper cells residing in the intestine wall [62,63]. It is not surprising then that CX3CR1-deficient mice’s gut characterizes increased bacterial translocation and intestinal inflammation mediated by IL-17 [64].

The loss of the aforementioned delicate balance allows the uncontrolled proliferation of many pathogenic bacteria which up to this moment were not allowed to thrive by the presence of functional microflora. That is the first step to the development of many diseases, starting from bacterial infections (where pathogenic bacteria cause e.g., diarrhoea), inflammatory bowel disease (IBD) and ending in metabolic syndrome and cancers.

## 5. *Enterococcus faecalis* Can Cause Life-Threatening Conditions

The ability to colonize intestines is one of the features allowing the bacteria to become a part of microflora, but if that ability is coupled with antibiotic resistance and the presence of specific, genetic factors, it enables the bacteria to become pathogenic. Many teams already presented extensively the virulence determinants of *Enterococcus faecalis*, e.g., Ferchcichi et al. [39] and Kayaoglu & Orstavik [65], so the present paper will not do it again, but some worth mentioning are aggregation substance (AS), adhesins, hyaluronidase, sex pheromones and gelatinase. All of them, but also others when combined with antibiotic resistance can pose a serious threat to the function of specific systems and the human organism as a whole. It is believed nowadays that *Enterococcus faecalis* is one of the most prevalent multidrug-resistant hospital pathogens worldwide [66].

The feature, which cannot be ignored when it comes to the pathogenic nature of the bacteria, is their ability to form biofilm. This is of course not unique to enterococci but it was already proved, that the problems with effective eradication of these bacteria are commonly linked with their ability to encase themselves in extracellular matrix containing e.g., polysaccharides, proteins and lipids. That in itself creates a virtually impossible-to-penetrate structure by both antibiotics and immune system cells decreasing the chance for successful elimination.

The pathogenic properties are acquired by the bacteria through the horizontal transfer of the pathogenicity and drug-resistance-associated genetic elements [67]. Generally, it is believed that the bacteria is especially dangerous in patients with already disturbed GI tract microbiome e.g., by extensively used antibiotics, as it allows for the selection of microbes with specific properties both affecting the drug resistance and the survival in the non-physiological environment. It grants the acquisition of new mobile genetic elements by those bacteria which do not own them already. It was established that *E. faecalis* is responsible for at least 11% of hospital-acquired infections (HAI), depending on the analysed statistics, with increasing morbidity and mortality throughout the years [68,69]. 

The first mentions of intestinal bacteria able to cause serious infections date back to the 19th century, and the first detailed enterococcus-caused infection comes from the paper dated 1899 describing enterococcal endocarditis [70]. It is assumed that nowadays *E. faecalis* may be responsible for up to 10–20% of community-acquired cases of endocarditis [15,71]. Apart from that, *Enterococcus faecalis* can play a vital role in the development of many other diseases, among them: intraabdominal infections, urinary tract infections (UTIs), prostatitis, wound infections and sepsis.

### 5.1. Enterococcus faecalis and Intraabdominal Infections

*E. faecalis* is one of the most common causative agents for intraabdominal infections (IAI). According to Luo et al., about 25% of infections in intensive care units (ICUs) are caused by those bacteria [72]. The general risk factors discussed when it comes to the IAI are as follows: abdominal surgery, including pancreaticoduodenectomy [73], and liver transplantation [74,75]. Especially in the case of liver transplantation, when the patient is immunocompromised it is important to start on antibiotics that cover enterococci according to Kajihara et al. [75]. Additionally, Morvan et al. proved in a retrospective study that when the initial antibiotic therapy does not cover enterococci IAI is linked with higher 30-day mortality than when the enterococci-specific antibiotics are used [76].

Kajfasz et al. claim that the successful *E. faecalis* growth and propagation in peritonitis, including the dissemination to the bloodstream, may depend among others on the presence of a Spx regulator [77]. They discovered that the Spx regulator is responsible for the ability of the bacteria to survive oxidative stress. Oxidative stress is the major mechanism exhibited by macrophages and neutrophils in so-called oxidative burst preceded by phagocytic engulfment. In theory, the production of NO (nitric oxide) and reactive oxygen species (ROS) should stop the growth of bacteria and cause their death, efficiently stopping the development of infection. Nowadays it is common knowledge that the ability of *Enterococcus faecalis* to survive oxidative stress is one of the major virulence attributes allowing among others the successful spreading from the physiological niche (e.g., GI tract) to other, non-physiological sites [78,79]. The resistance to intra-phagosomal killing is not only based on the blockade of ROS formation. It can also be utilized by low pH resistance. As was mentioned in the introduction, *Enterococcus faecalis* is resistant to low pH as it helps them to survive in the GI tract, but from the ‘pathogenic’ point of view, the resistance to the low pH means also the ability to resist phagosome acidification and effective autophagy [80].

Another interesting feature is the ability of *E. faecalis* to generate ROS by themselves. That in itself helps this facultatively anaerobic bacteria to survive in the presence of oxygen but is also believed to be one of the mechanisms allowing the spreading of the bacteria throughout the body of the host by damaging the DNA of the cells and inducing carcinogenesis [81]. According to Leger et al., the ability of bacteria to produce ROS can be stimulated by specific antibiotics (e.g., beta-lactams—amoxicillin) which needs to be taken into consideration as another mechanism debilitating for the patient—in regards to e.g., colon cancer development and dysbiosis formation [82]. Riboulet et al. presented a nice summary of enterococcal genes and proteins involved in enterococcal oxidative stress response [78]. 

Enterococcal surface protein (Esp) is another one, which plays an important role in the development of peritonitis, among other infections. It is typically associated with highly virulent strains of enterococci as it is isolated mainly from infection-derived strains rather than from commensal ones [83]. It was proved by Zou et al. that Esp can induce the production and release of proinflammatory cytokines, such as TNF-α, IL-6 and MCP-1 (monocyte chemoattractant protein-1) via the induction of NF-κB pathway in neutrophils. The levels of proinflammatory cytokines in mice peritoneal fluids were also much higher and the infiltration of the liver by neutrophils was more prominent when the animals were infected with Esp-positive enterococcal strains than Esp-deficient strains [84].

The main immune mechanism regulating the response to bacterial offenders is the recognition of conservative PAMPs by PRRs as mentioned earlier. The data from *Enterococcus faecium* peritonitis suggest that the element maintaining the immune response against enterococci is mainly TLR2 as it recognizes LTA and lipoproteins. Together with MyD88 it activates peritoneal macrophages and their shift towards the M1 phenotype [84,85,86], recruits the neutrophils to the primary site of infection [87], and stimulates the opsonisation of bacteria by the complement system [88] ensuring the clearance of the bacteria [89]. In the case of intraabdominal abscess formation, which is considered an important step in the development of either sepsis or an infection chronic in nature, the adaptive immune system plays its role. It was proved that T cell activation by APCs through the CD28-B7 interaction is crucial for abscess formation as well as that the signalling through CTLA-4 (which is a cellular antagonist of CD28)-B7 inhibits abscess formation [90].

### 5.2. Enterococcus faecalis and Urinary Tract Infections

Urinary tract infections are quite commonly caused by *Enterococcus faecalis*. In reality, it is the most common gram-positive bacteria causing the problem. The risk factors for those infections are numerous including congenital and acquired structural abnormalities of urinary tract elements, male sex, stay in ICUs, hypoalbuminemia, and the use of broad-spectrum antibiotics, as well as artificial devices such as nephrostomies and stents [91]. But truth be told, they are not specific for enterococcal infections, because all those factors are considered risky in any kind of bacterial infection. 

In previous years the role of *Enterococcus faecalis* in the development of UTI seems to be increasing. The increasing multi-drug resistance, the ability to form biofilm and the natural ability to grow and spread in the urinary tract make them especially hard to eradicate. According to Horsley et al., another feature of *E. faecalis* that makes it such a formidable bacteria is its ability to invade uroepithelium [92]. The host’s immune reaction to the bacterial invasion of the urinary tract is not surprisingly, the infiltration with white blood cells (WBC) (pyuria). Another—innate mechanism—is the shedding of urothelial cells into the urine. Surprisingly, *E. faecalis* but not *Escherichia coli* can invade the shed cells becoming resistant to virtually every antibiotic [92].

The bacteria itself shows the greatest tropism for the kidneys when it comes to the urinary tract, but can cause an infection in all the elements of the system [93,94]. The pathogenicity of *E. faecalis* in the context of the urinary tract is linked with many genetic features and it is worth to underline, that those features rarely are linked with one specific type of infection or condition. 

One of the loci worth mentioning in the context of UTIs (but also in the case of peritonitis or endocarditis) is enterococcal polysaccharide antigen (Epa) as it allows for the binding to epithelial cells and biofilm formation and/or resistance to phagocytosis [75,91]. The ability to create biofilm and resistance to phagocytosis (especially in the case of bacteria in the form of biofilm) seems to be especially important and dependent on the *epa* gene cluster as the mutants are more susceptible to the neutrophils-mediated phagocytosis, and have a lower ability to form a biofilm, therefore, are more effectively cleared from the organ [95,96]. The presence of Epa locus is also linked with increased resistance to ceftriaxone and carbapenems [97]. 

Ebp (endocarditis and biofilm-associated pili) is another element which facilitates the interaction between the bacteria and urinary tract epithelium. According to Singh et al. as well as Nallapareddy et al. *ebp* locus is responsible for the colonization of the kidneys and bladder in the urinary tract infection model [94,98].

Statistically, most UTIs are acute and the diagnosis of those is not considered difficult due to their rapid onset and specific set of symptoms, but even that can become finicky in a specific group of patients, whose immune system is somehow challenged, such as immunocompromised patients due to e.g., renal transplantation (RTx). It is assumed that their muted immune response links with rather non-specific (or even not present at all) symptoms. The presence of immunosuppression also causes the selection of specific enterococcal strains. Increased pheromone cCF10 (cell-associated pheromone peptide 10) expression is typical for biofilm-forming isolates from RTx patients’ urine [99]. AS (aggregation substance) gene expression is much lower in biofilm-forming urine bacteria isolated from immunocompromised patients with ADPKD (autosomal dominant polycystic kidney disease), and glomerulonephritis than in healthy peoples’ faeces-isolated biofilm-forming strains [100]. The expression of PBP5 (penicillin-binding protein 5) is dependent on the used immunosuppressive drug as it is higher when the cyclosporine is used rather than tacrolimus [101]. Another thing is the existence of so-called small colony variants (SCVs) of *Enterococcus faecalis*. Those bacteria, characterised by low proliferation rate are much more resistant to antibiotic action, e.g., penicillin. They also tend to cause chronic, hard-to-eradicate infections as they are characterised by higher expression of surface adhesins than typical variants of bacteria [102].

As mentioned above, enterococcal infection activates the innate immune system through its ability to recognise PAMPs (LTA and lipoproteins in the case of *E. faecalis*). Dendritic cells which recognise PAMPs, engulf the antigens and present them to adaptive immune cells, namely T lymphocytes. According to Kathirvel et al., dendritic cells can be infected by enterococcal cells and the infection is a crucial step for the activation of NK cells, which upon activation can release proinflammatory cytokines, such as IFN-γ [103]. As most of the UTIs are poly-microbial, the effect of another bacteria co-existence in the infected niche needs to be taken into consideration. Tien et al. have shown that catheter-associated urinary tract mono-infection with *Enterococcus faecalis* prevents macrophages’ NF-κB signalling inhibiting that way the pro-inflammatory cytokines and chemokines production. The co-infection with commensal, non-virulent *E. coli* revealed that *E. faecalis* limits *E. coli*-mediated immune activation and therefore promotes this bacteria’s (so far non-existing) virulence [104]. Kao et al. on the other hand reported that during the co-infection with *Enterococcus faecalis* and *Staphylococcus aureus*, *E. faecalis* can suppress NETs (neutrophil extracellular traps) formation and therefore protect *S. aureus* from the killing [105]. NETs normally allow the innate immune system for the clearance of bacteria and *S. aureus* as opposed to *E. faecalis* is sensitive to this form of elimination [106,107]. 

### 5.3. Endocarditis Caused by Enterococcus faecalis Strains

Infectious endocarditis (IE) epidemiology and microbiology have changed in recent years and the main change is the increasing role of enterococcal infections. Nowadays it is believed that enterococci are the second causative agent for IE, directly after staphylococci [108]. The change is explained by the general population ageing. The changes observed in older people such as changes in lifestyle, mobility and diet as well as immune system functioning and linked with that, recurrent infections, hospitalizations and the use of drugs influence dysbiosis development and the selection of bacterial strains in the gut [109,110,111]. Adding to that changes in the gut wall anatomy and physiology, higher is also the risk for bacteria leakage to the systemic circulation. As enterococci’s ability to cross the colon mucosa is well known, that in turn may explain the increasing role of enterococci in the development of IE [60,111,112]. It was proved by Panteris et al. that 88% of patients with enterococcal IE had also colorectal disease [113].

As in the case of UTI, the ability of *E. faecalis* to cause endocarditis is partially dependent on the presence of Epa. As mentioned above, Epa allows for biofilm formation and decreases the effectiveness of phagocytic clearance [95,96]. Additionally, Schlievert et al. showed that both AS and Ebs (enterococcal binding substance) are needed for fatal endocarditis development. In their experiment, rabbits infected with AS(-)Ebs(-) *E. faecalis* did not develop endocarditis, AS(+)Ebs(-) strains injection led to the development of the endocarditis symptoms but all rabbits survived. The rabbits infected with AS(-)Ebs(+) strains developed endocarditis, and 10% of them died due to the disease. The endocarditis caused by the infection with AS(+)Ebs(+) *E. faecalis* strain led to the death of all tested animals [114]. The same team assessed also the response of human T lymphocytes to the AS(+)Ebs(+) bacteria as well as AS(-)Ebs(-) bacteria. The contact with the first type of bacteria led to the activation and proliferation of T cells and subsequently to the production and release of IFN-γ, and TNF-α cytokines as well as the activation of macrophages and TGF-β production by them. The exposure of T lymphocytes to AS(-)Ebs(-) bacteria failed to activate lymphocytes at all [114].

## 6. *Enterococcus faecalis*—Human Immune System Interplay

Table 2 summarizes the wide variety of mechanisms used by *E. faecalis* to hide from, or deceive the human immune system cells. As the main identified so far mechanisms take into consideration the innate immune system only, adaptive immune system elements were not included in the table.

## 7. Conclusions

*Enterococcus faecalis* is a multifaced bacteria. ne hand, it can be a commensal with a positive impact on the functioning of GI tract microflora (among others) and quite an effective probiotic. On the other side, it can be responsible for many highly complicated infections, especially in those already immunocompromised by underlying conditions and their treatment or age. As mentioned above the bacterium is one of the most common nosocomial pathogens nowadays with increasing drug resistance which complicates successful treatment. Even though the presented paper did not cover that, it is worth mentioning that it is believed that the opportunistic nature of the bacteria is also caused by the common usage of bacteria in the food industry, especially in dairy products and fermentation. The widespread use of *E. faecalis* in industry raises a question about its safety, especially if we take into consideration the increasing cunning of the bacteria in omitting the human immune system, leading to a decreasing rate of successful eradication of the bacteria.

## Figures and Tables

**Table 1 ijms-25-02422-t001:** *Enterococcus faecalis* strains which are used as probiotics for humans (based on [38,39], changed).

The Name	Enterococcal Strain	Other Bacteria Present in the Formulation	The States in which the Supplement Was Tested and/or Used
Bifilac	*E. faecalis* T-110 (named as *Streptococcus faecalis* T-110)	*Lactobacillus sporogenes*, *Clostridium butyricum TO-A*, *Bacillus mesentericus TO-A JPC*,	− diarrhoea of various origins [40],
Bioflora	*E. faecalis* (named as *Streptococcus faecalis*)	*Lactobacillus casei*, *L. plantarum*, *Bifidobacterium brevis*,	− intestinal homeostasis restoration after antibiotic or chemotherapy, candidiasis, SIBO [41],
BIO-THREE	*E. faecalis* T-110 *	*Clostridium butyricum TO-A*, *Bifidobacterium mesentericus TO-A*,	− intestinal homeostasis regulation, − ulcerative colitis, − prevention of colon cancer, − acute diarrhoea [42],
ProSymbioflor	*E. faecalis* DSM 16440	*Escherichia coli* DSM 17252,	− irritable bowel disease [43], − to reduce the risk for atopic dermatitis in predisposed patients [44,45],
Shin-Biofermin S	*E. faecalis* ** (named as *Streptococcus faecalis*)	*Bifidobacterium bifidum*, *Lactobacillus acidophilus*,	− intestinal homeostasis regulation, − diarrhoea treatment,
Symbioflor-1	*E. faecalis* DSM 16440	none	− chronic bronchitis [25], − recurrent rhinosinusitis in children [46,47], − seasonal allergic rhinitis [48], − asthma in children [49],
ThreeLac (x)/ FiveLac (y)/ SevenLac (z)	*E. faecalis*	*Bacillus coagulans* (x/y/z), *B. subtilis* (x/y/z), *Bifidobacterium longum* (y/z), *Lactobacillus acidophilus* (y/z), *L. rhamnosus* (z), *L. johnsoni* (z),	− intestinal homeostasis regulation, − candidiasis,

* The manufacturer claims it to be *Enterococcus faecium*, Natarajan et al. assessed the genome sequence of bacteria and it was deemed *Enterococcus faecalis* though [50]; ** According to Gan et al. the bacteria present in the probiotic is *Enterococcus faecium* based on the complete genome sequence [51]; SIBO—small intestinal bacterial growth.

**Table 2 ijms-25-02422-t002:** The immunological mechanisms, which may be affected by pathogenic *Enterococcus faecalis* (based on [107], changed).

Immunological Mechanism	Immune Cells taking Part in the Process	*Enterococcus faecalis* Mode of Action	Possible Role in the Pathologies Development	References
Oxidative burst in the phagocytosis process	Macrophages and granulocytes	− the ability to inhibit caspase 3 and activate the PI3K/Akt signalling pathway to delay apoptosis of infected cells,	− spreading of the bacteria throughout the body,	[80,115]
Phagolysosomes organisation in the phagocytosis process	Macrophages and granulocytes	− resistance to acidification of the environment and low pH,		[80]
Autophagy/xenophagy	Macrophages and granulocytes	− the ability to increase ROS production and inhibit autophagy-related proteins, such as LC3,	− the blockade of phagocytosis and as a result spreading of the bacteria throughout the body,	[80]
	Dendritic cells	− the inhibition of the production of the proteins related to autophagy, such as beclin-1 and LC3,	− the periapical lesions’ development,	[104]
Cytokines production	Macrophages and granulocytes	− the inhibition of NFκB activation and the suppression of the inflammatory cytokines production,	− the development of chronic infections,	[116]
	Dendritic cells	− the downregulation of cells’ maturation and activation of pro-inflammatory cytokines’ genes expression	− the tissue damage and bone resorption in chronic apical periodontitis, − the periapical granulomas formation in primary teeth,	[104,117]
Opsonisation	Granulocytes	− the adhesion to the surface of granulocytes mediated by AS, therefore resistant to opsonisation	− the development of chronic infections,	[118]
NETosis	Granulocytes	− the prevention of NETosis in mixed infections,		[105]

## Data Availability

Data are contained within the article.

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
