# Peer review of "From the Friend to the Foe—Enterococcus faecalis Diverse Impact on the Human Immune System"

_ijms, 2024, doi:10.3390/ijms25042422_

Round 1

Reviewer 1 Report

Comments and Suggestions for Authors

 The authors showed the role of E. faecalis and their positive effect on the human immune system when used as a probiotic. Then, they explain the negative and diverse effects of E. faecalis on immunocompromised patients and the ability of these bacteria to cause serious diseases.

Comments:

1-      The mechanisms by which E. faecalis can cause infections were not addressed well in the manuscript. Authors should explain more and discuss the mechanisms and modes of action to cause infections.

2-      The authors focused on three major diseases (intraabdominal infections, urinary tract infections, and endocarditis). What about other serious diseases like Enterococcal bloodstream infections which can be caused mainly translocation of enterococci from the gut into the bloodstream

3-      The authors should underline the negative impact of eating food contaminated with this pathogen. E. faecalis is an opportunist pathogen so they only affect immunocompromised patients.

4-      The ability of E. faecalis to form biofilm should be taken into consideration. The relation between biofilm formation and causes of diseases should be discussed.

Reviewer 2 Report

Comments and Suggestions for Authors

1. Table 1 which is referenced to show Enterococcus faecalis used as supplement - Year of publication and year in which such formulation was recommended for supplementing in humans need to be incorporated in table

2 A paragraph on fecal transplant with Enterococcus fecalis may be added if available in literature . 

3  Pathogenic genes with a horizontal gene transfer in strains like DSM 16440 is possible or not - please comment upon this issue 

4. How enterococcus fecalis help in gut brain axis . Is this a universal phenomenon seen in all humans ? 

5 Is enterococcus fecalis is seen as a commensal bacteria in all humans .? 

6 Review should be incorporated with some graphic pictures and appropraite tables to make it intresting to read and understandable  (Suggestion) 

Reviewer 3 Report

Comments and Suggestions for Authors

The manuscript "From the Friend to the Foe – Enterococcus faecalis diverse impact on the human immune system" is a nice summary of the topic focusing on immunological aspect of the E. faecalis and the human body. My opinion is that  you should mention more visible that this species is very often part of mixed infections and because of this many aspects discuss here may be influenced by co-pathogens. It might be also mentioned that antibiotic therapy of mixed infections where E. faecalis is involved my be influenced by its ability to prevent other bacteria (anaerobes) from the activities of otherwise active drugs e.g. metronidazole.

Many of my suggested corrections are included to the attached manuscript

Special comments: based on the requirement of the journal you may use first the hole name of the bacterium Enterococcus faecalis otherwise only E. faecalis or if the requirement to write out the genus name everywhere Please do it as the rule of the journal that but not in a mixed way.

Basic English correction is needed. "Bacteria" is prular, "bacterium " is sigle. It is mixed up in many places. 

Comments on the Quality of English Language

Corrections are needed See the attached file
